# Solder Joint Reliability Risk Estimation by AI-Assisted Simulation Framework with Genetic Algorithm to Optimize the Initial Parameters for AI Models

**DOI:** 10.3390/ma14174835

**Published:** 2021-08-26

**Authors:** Cadmus Yuan, Xuejun Fan, Gouqi Zhang

**Affiliations:** 1Department of Mechanical and Computer-Aided Engineering, Feng Chia University, Taichung 407082, Taiwan; 2Department of Mechanical Engineering, Lamar University, Beaumont, TX 77710, USA; xuejun.fan@lamar.edu; 3Department of Electronic Components, Technology, and Materials, Delft University of Technology, 2628 CD Delft, The Netherlands; g.q.zhang@tudelft.nl

**Keywords:** solder joint fatigue risk estimation, wafer level chip-scaled packaging, artificial neural network, recurrent neural network, generic algorithm, principle component analysis, time/temperature-dependent nonlinearity

## Abstract

Solder joint fatigue is one of the critical failure modes in ball-grid array packaging. Because the reliability test is time-consuming and geometrical/material nonlinearities are required for the physics-driven model, the AI-assisted simulation framework is developed to establish the risk estimation capability against the design and process parameters. Due to the time-dependent and nonlinear characteristics of the solder joint fatigue failure, this research follows the AI-assisted simulation framework and builds the non-sequential artificial neural network (ANN) and sequential recurrent neural network (RNN) architectures. Both are investigated to understand their capability of abstracting the time-dependent solder joint fatigue knowledge from the dataset. Moreover, this research applies the genetic algorithm (GA) optimization to decrease the influence of the initial guessings, including the weightings and bias of the neural network architectures. In this research, two GA optimizers are developed, including the “back-to-original” and “progressing” ones. Moreover, we apply the principal component analysis (PCA) to the GA optimization results to obtain the PCA gene. The prediction error of all neural network models is within 0.15% under GA optimized PCA gene. There is no clear statistical evidence that RNN is better than ANN in the wafer level chip-scaled packaging (WLCSP) solder joint reliability risk estimation when the GA optimizer is applied to minimize the impact of the initial AI model. Hence, a stable optimization with a broad design domain can be realized by an ANN model with a faster training speed than RNN, even though solder fatigue is a time-dependent mechanical behavior.

## 1. Introduction

Solder joint reliability is one of the most critical issues for most ball-grid array packaging types. The time dependency of this failure mechanism requires considerable experiment time to obtain statistically reliable results. On the other hand, the nonlinear material/geometry properties are required for the finite element (FE) modeling to retrieve trustable results, which can be validated by the experimental results. Hence, both the reliability experiment and numerical modeling require unique expertise to conduct the relevant tasks, which creates a technical barrier for the design for reliability.

The neural network (NN)-based AI algorithms were applied to assist the design and simulation of the solder joint risk assessment. Chou and Chiang [1] and Hsiao and Chiang [2] developed an AI-assisted design and simulation framework. It includes the virtual prototyping of solder fatigue failure mode with the geometrical/material nonlinearity and the proper validation by the experimental results. The training database is generated from the parametric FE model. Next, AI modeling is trained by the selected data points from the database and validated by the rest. Careful validation works should be conducted to prove the representation capability of the AI model to the FE dataset.

The NN approaches of AI modeling received more and more attention due to the capability of abstracting the knowledge from the database without the pre-defined framework nor prior knowledge/experience. ANN is a basic architecture of NN with weak representation capability for sequential events. However, the solder joint fatigue mechanism is highly time-dependent. Accordingly, the sequential NN techniques, including the recurrent neural network (RNN), GRU, and LSTM, are successfully applied to the time-dependent failure mechanism for electronic packaging. However, these sequential NN methods consider the recurrent parameters and iterations that induce learning difficulties and require considerable computation resources. Yuan and Lee [3] applied the sequential NN to model the time-dependent nature of the solder joint fatigue, and the average error norms below 1.213 × 10^−4^ were achieved. Yuan et al. [4] developed a gated neural network technique to learn the performance shifting of the solid-state lighting (SSL) lamp over time. Meszmer et al. [5] applied many NN techniques to study which is the best for the electronic packaging, and the sequential NN performed best, including the gate recurrent unit (GRU) and long short-term memory (LSTM) architectures because of their capability to capture the characteristics of the sequential dataset. Selvanayagam et al. [6] applied the AI-assisted modeling concept for the improvement of the packaging warpage. Tabaza et al. [7] applied the non-sequential NN to simulate the time-dependent hysteretic response of a viscoelastic material. The possibility of using the non-sequential NN, such as NN, upon a time-dependent engineering problem remains a challenge.

Considering the learning procedure, NN is a parametric AI modeling method. In addition to the network structure, NN utilizes the parameter, including the weightings and bias, to learn the knowledge from the database. However, NN requires the initial guessing of parameters to start the learning process, and the improper selection of the initial parameter results in a slow convergence speed and bumpy learning procedure. The genetic algorithm (GA) is always applied. A genetic algorithm (GA) is an optimization method proposed by John Holland to find the approximate solutions. This algorithm is a specific form of an evolutionary algorithm in which evolutionary biology techniques such as inheritance and mutation are used. In genetic algorithms, to obtain the optimal solution, the appropriate responses of a generation are combined based on the principle of the survival of the fittest environment [8]. White and Ligomenides proposed a topology and weighting optimization algorithm for neural networks [9]. Juang [10] hybrids the GA and particle swarm optimization (PSO). The new generation of the GA can be generated not only by the crossover and mutation but also by PSO. Ding et al. [11], Ahmadizar et al. [12] and Arabasadi et al. [13] apply the GA for weighting optimization of NN. However, few literature had covered GA for the sequential network nor utilized the continuously evolving nature of the NN backpropagation.

Based on the AI-assisted simulation framework, this research investigates the possibility of using the ANN instead of the RNN in physically sequential issues, such as the solder joint fatigue mechanisms. Secondly, this research develops a genetic algorithm method to obtain the optimal parameters for NN learning.

This paper is organized as follows: the fundamental scientific issues and the literature review are described in the Introduction. The following section, Theory, provides the basic theoretic approaches that have been applied in this research. The execution of the AI-assisted simulation framework is explained in the sections The AI-assisted simulation framework and FE dataset preparation and The design of the AI modeling. The AI model training with GA optimized initial parameters section summarizes the learning experience of the AI modeling. The conclusion of this paper is given in the last section.

## 2. Theory

In GA, the fitness criterion is first defined to quantify the members of the current generation with more compatibility are more likely to generate the next population [8,9]. The fitness criterion (F) is set as follows: (1)F=1/r
where the r is the error norm of learning. The members with higher F values are more likely to generate the next population by the crossover and mutation operators.

The member is also called the chromosome, which is made up of many genes. The gene is constructed by many base-pairs (bps) [12]. Given a pair of parent chromosomes with m genes, the crossover operator will generate $2^{m}$ different offsprings by the recombination of the genes that are from the parent [13]. To cover the genetic representation ultimately, the whole $2^{m}$ offsprings are forming to the next generation without any possibility. As illustrated in Figure 1, there is one pair of parents with three genes. Eight ($=2^{3}$) offsprings are generated by the recombination of the parents’ gene, which is the definition of the crossover operator.

Therefore, if the top n chromosomes are selected, there are $\frac{n\left(n-1\right)}{2}$ parent combinations and there are maximum $\frac{n\left(n-1\right)}{2}\times 2^{m}$ offsprings possibilities. However, if the parent chromosomes consisted of many similar genes, it might induce many duplicated offsprings. These offsprings with the same genes will be removed to save the computation resource.

The mutation operator [13] is used to make changes in the genes of a member of the current generation to produce a new member. The mutation occurs at the bps level and is controlled by the mutation rate [12]. When the mutation is invoked at certain bps, the representation bps will be replaced by the opposite parent bps. For example, as illustrated in Figure 1 and Figure 2, the 9th bps of the first gene of offspring 7 will mutate, and the original bps will be replaced by the 9th bps of the first gene of parent 1 (P1).

After few completed genetic algorithm iterations, these best chromosomes of each iteration are not the same due to the nature of the neural network. The principal component analysis (PCA) algorithm is applied to build a super chromosome based on these best chromosomes. If the complete chromosome, including the weightings and bias, are considered as vectors, and the covariance matrix K of all the chromosomes are formed based on the squared exponential kernel function:(2)$$K\left(x,x^{\prime }\right)=\sigma \cdot \exp\left(-\frac{{∥x-x^{\prime }∥}^{2}}{2l^{2}}\right)$$
where the x and x′ are the chromosome vectors. The parameter σ and the characteristic length l are both set as 1. An eigenvector analysis is applied to the K matrix. The super chromosome, called the PCA gene, is obtained as the inner product of the best chromosomes and the eigenvector of the first eigenvalue.

## 3. The AI-Assisted Simulation Framework and FE Datasets Preparation

In this section, a practical engineering case will be applied to analyze the capability of GA and PCA to generate the initial parameters of the ANN and RNN. However, due to the multiphysical and multiscaled characteristics of the engineering questions, this research obeys a reliable AI-assisted simulation framework [1,2,3], illustrated in Figure 3, to improve the predicting accuracy of the AI model.

Due to the limited resources, only limited actual samples with very few design parameter combinations are made for the experiments. The FE modeling method is applied to expand the design domain. Based on the experimental results (Figure 3a), a FE model (Figure 3b) can be established and validated. The validated FE model then can be parameterized, and the FE datasets (Figure 3c) can be obtained. However, the specified design parameter combination might induce bad aspect ratios of the elements, which cause the instability of the FE analysis results. The NN model is expected to broaden the design domain. The NN will be carefully designed (Figure 3d) based on the characteristics of the FE dataset and supervised trained (Figure 3e). Moreover, the AI model training’s accuracy requirement(s) should be carefully defined based on the FE dataset. The datapairs that have not been included in the training procedure will be applied to validate the NN model. The validated NN model can be used for design and optimization (Figure 3f). When the new experimental result is available (Figure 3g), the whole procedure can be relaunched.

Figure 4 shows the G-WLCSP structure [14], where the IC is placed on a glass substrate with metal traces and solder bumps Figure 4a for redistribution purposes. Figure 4a’,b show the device’s top view and cross-section view, respectively. Moreover, Figure 4b depicts the key structural components in G-WLCSP, including the glass substrate, the adhesive, the IC, polyimide (PI) for the stress buffer layer, solder mask, and the solder. After the wafer has been diced, individual packaging can be obtained.

An actual G-WLCSP structure was made for reliability testing. The sample consisted of a chip with the size of 5.77 × 10.38 × 0.3 mm^3^ and a glass thickness of 0.5 mm. The sample was attached to a 1.2 mm-thick test board, as shown in Figure 5a. The cross-section view shows the bonding condition, and no defect has been detected (Figure 5b). A 0.45 mm-diameter 63Sn/37Pb solder ball was applied onto the 0.37 mm die-side pad. The stand-off height for solder joints was reduced to 0.35 mm after reflow.

Thermal cycle testing of this G-WLCSP is performed between −40 °C and 125 °C with a ramp rate of 11 °C/min and a dwelling time of 15 min. Figure 5c shows the Weibull solder fatigue failure experimental result of 21 samples; the 63.2% fatigue cycle number is approximately 1444.

We develop a two-dimensional FE model with a plane strain assumption to estimate the G-WLCSP solder joint risk under the thermal cycle loading by the incremental plastic strain. The initial stress-free reference temperature equals 25 °C. In the finite element model, all materials except the solder joint and the PI are linear, as shown in Table 1. Moreover, the solder joints and PI are treated as temperature-dependent, elastic-plastic materials [15,16], as shown in Figure 6 and Figure 7, respectively. As seen in Liu et al. [16], the solder joint failure risk can be estimated at a certain accuracy level without the time-dependent material properties. Due to the symmetrical condition, one-half of the full-scaled two-dimensional FE model is used, and the analysis result is obtained by the commercial finite element code ANSYS^®^ (version 15, ANSYS, Inc., Canonsburg, PA, USA). The mesh density of the most critical solder is shown in Figure 8.

By fine-tuning the mesh density and the solution parameters, the FE model can achieve good agreement with the experimental results by the empirical Coffin–Mason equation [17,18,19], shown in Table 2. Afterward, the validated FE with the solution parameters can be parametrized.

Key parameters with the levels and noise factors are listed in Table 3. These three key design parameters have been chosen in response to the packaging industry requirements and manufacturing capabilities. A finite element model based on experimental validation is first used to broaden the domain of parameters, and then the neural network model is applied. Each simulation comprises a complete five thermal cyclic loading. To build the FE dataset, 81 parametric finite element models, according to the parameters in Table 3, are executed with a controlled mesh density of the most critical solder joint.

Figure 9 shows the averaged incremental plasic strain of each loading cycle, where the plastic strain is only induced by plasticity deformation. After the third cycle, the averaged incremental plastic strain incremental becomes stable. From these 81 data points, the average strain increment is 3.01% ($\Delta \epsilon_{avg}$), with a standard deviation of 1.17% ($\sigma_{\epsilon }$). The empirical Coffin–Mason equation [17,18,19] converts to the reliability cycles, as
(3)$$N_{f}\left(\Delta \varepsilon_{p}\right)=0.4405\cdot {\left(\Delta \varepsilon_{p}\right)}^{-1.96}$$

On the other hand, Table 2 indicates that the difference between the experimental and simulation result is 437 cycles. Based on Equation (3), we define a max-min problem: (4)f=arg⏟Δεaccu∈ℝ{min[max(ΔNf−Δ)]}
where ΔNf is defined as |Nf(ϵ+Δϵaccu)−Nf(ϵ−Δϵaccu)|,  ϵ=N(Δϵavg,σϵ), and N(·,·) represents the Gaussian distribution. The |f| in Equation (4) is expected to be zero. By the regression computation, Δϵaccu~0.18% is obtained, and it is assigned as the accuracy requirement for machine learning.

## 4. The Design of the AI Model

Figure 10 schematically illustrates the characteristics of the datapairs. There are three design parameter inputs, including the die, glass, and PI thickness. Since the FE model is under five thermal cycle loadings, denoted as the cycle 1–5 illustrated in Figure 10. At each cyclic thermal loading, there is an equivalent plastic strain ($\Delta \epsilon_{pl}$) with respect to each temperature.

Hence, the ANN structure is designed as follows: the three geometric design parameters are considered as the inputs. Referring to the plastic strain incremental of the 81 data points shown in Figure 9, the average equivalent plastic strain increment of the last three loading cycles is selected as the output. The design concept of ANN is to capture the relationship directly from the design parameters to the solder joint fatigue cycle, which is represented by the equivalent plastic strain and converted by the Coffin-Manson equation (Equation (3)).

The ANN structure is designed as “3,4,4,1”. There are three inputs and one output, and there are two hidden layers, including the weightings and bias, to capture the feature characteristics of the training datasets. The sigmoid is selected as the activation function because it is stable for the initial parameter studies in the next paragraph. To keep the simplicity during learning, the ANN optimizer is limited to forward computation and backpropagation. The data normalization is applied to the datasets.

Each prediction error is defined as e=o−t, where o is the prediction obtained from the ANN output, and t is the ground truth from the FE datasets. The cost function is defined as the Euclidian sum of each prediction error: (5)$$C=\sqrt{\frac{\sum_{i=1}^{n}e_{i}}{n}}$$
where $e_{i}$ is the prediction error of $i^{th}$ prediction and there are total n predictions.

On the other hand, the plastic strain accumulated from the previous cycle will impact the system’s mechanical response. Hence, RNN is applied to predict the equivalent plastic strain at each cycle. The equivalent plastic strain (per cycle) is the output. Moreover, there are five inputs, including the three geometrical design parameters, the temperature of the current cycle, and one recurrent parameter. Referring to the top-right schematic drawing in Figure 3, the recurrent parameter of RNN converts the previous output of equivalent plastic strain into the input of the next cycle.

The structure of RNN is set to “5,4,3,1”. There are five outputs and one input, and two hidden layers with four and three neurons, respectively. The design concept of the hidden layer is to keep similar numbers of weightings and bias, to compare to the ANN results directly. The sigmoid is selected as the activation function. To maintain the simplicity during learning, the ANN optimizer is limited to forward computation and backpropagation through time. The data normalization is applied to the datasets. One extra post-processing is applied to the ANN. Only the average of the last three outputs is considered to compare to the ANN prediction accuracy directly.

## 5. AI Model Training with GA Optimized Initial Parameters

In neural network-based AI modeling, the initial parameters are required to launch the machine learning process. This initial parameter is usually chosen randomly in the literature. In this paper, a GA is proposed to achieve the best initial parameter for the AI modeling. The GA chromosome is defined as the combination of genes. Each gene is the combination of the weightings or bias between two layers [12]. For instance, if there are two layers with $l_{m-1}$ and $l_{m}$ neurons, the gene occupies $l_{m}\cdot l_{m-1}$ individuals as illustrated in Figure 11. Each individual in the gene is defined as a base-pair (bps). The chromosome is a combination of genes from various layers.

The GA optimization procedure is illustrated in Figure 12. Each GA step starts with the old population of many chromosomes (Ch). The fitness ranking by Equation (1) limits that only m the best chrmosomes can enter the next population. By the crossover and mutation operators, a new population can be generated. The backpropagation of NN learning provides new fitness rankings for the next GA step.

In the AI-learning algorithm, the parameters, including the weightings and bias, are updated each iteration by the backpropagation process. Hence, there is an option to select which parameter sets generate the next GA generation. In this research study, two conditions are considered: 

“back-to-original” condition: only the parameters that initially input to the AI model are applied to generate the next generation.“progressing” condition: the parameters that after n backpropagation iterations are applied.

### 5.1. The “Back-to-Original” GA Optimizer

Due to the design of the neural network structure, 2000 initial parameter sets are firstly generated by the random generator, which follows a zero-mean Gaussian distribution of the standard deviation of $\sqrt{2/(l_{m}\cdot l_{m+1})}$ (0≤m≤n), where n is the total layer number, and $l_{m}$ is the neuron numbers at $m^{th}$ layer. It also indicates that 2000 initial chromosomes have been generated for each case to initialize the GA optimization procedure. The learning rate of ANN is fixed to 0.3 with the sigmoid activation function, and backpropagation is selected as the learning optimizer.

When the RNN architecture is applied with the “back-to-original” GA optimizer, Figure 13 shows convergence curves among random select chromosomes from the 2000 members (random selection), the best chromosome from the first generation (generation 1 best), and the best one from the whole GA optimization (GA best). A clear contribution of the GA optimization to the convergence speed can be confirmed.

A typical performance of the GA generations under the “back-to-original” optimizer is illustrated in Figure 14, where the lightest grey curve is best of the 2000 initials, and it evolves continuously from the light grey to the darker ones by the GA. The mutation rate for generating the next generation is fixed by 0.001. After the seventh generation, the evolving of the best chromosome stops. The inset of Figure 14 shows the average error norm among the GA generation and generation size of the GA. The average error norms reduce through each generation. The size of GA generation reduces accordingly as the duplicated chromosomes in the generation are removed. Moreover, the standard derivation of average norm varies during the GA optimization, but it dramatically decreases when the GA evolving stops (GES). This is because the optimized chromosome dominates the next GA generation.

We conducted four GA optimization procedures and listed them in Table 4, where the “Run-time error norm” column indicates the error norm of the best chromosome against the training sets after the 500 cycle training and before the denormalization.

Case 1 and 2 introduce the four best chromosomes to the next generation, and Case 3 and 4 introduce six. Due to that, the initial 2000 members were generated independently by the random number generator for each case in Table 4, a different optimization result is achieved under the same learning parameters. This phenomenon is clearly depicted in column “Run-time error norm” of Table 4.

After the GA optimization procedure, a PCA process is applied based on these four best chromosomes. The covariance matrix K is formed by Equation (2). An eigenvector analysis is applied to the K matrix. Then, The eigenvector of the first eigenvalue is selected and applied to those four best chromosomes. The PCA gene can be obtained by
(6)$$\left[ch_{1},\;ch_{2},ch_{3},ch_{4}\right]\cdot \vec{v_{e}}$$
where $ch_{1}$, $ch_{2}$, $ch_{3}$, and $ch_{4}$ are the best chromosomes from the GA optimization, and $\vec{v_{e}}$ is the eigenvector of the first eigenvalue. The convergence curve is shown in Figure 15, where a relatively fast convergence can be achieved. Moreover, from the inset of Figure 15, which extended the convergence curve to 100k iteration, one can identify that these best chromosomes approach similar error norm levels after considerable iteration numbers.

The “Denormalized difference” column in Table 4 indicates the difference between the best chromosome against the whole 81 datasets after 100,000 training cycles and after the denormalization process. Moreover, no matter the “Run-time error norm”, the best chromosomes can consistently achieve the 0.18% requirement. Moreover, when more best chromosomes are put into the next generation, the denormalized difference improves.

Under the similar GA condition of “back-to-original”, the RNN architecture has been implemented. Each GA generation is optimized by 1000 backpropagation through time iterations. The rest of the learning parameters remain the same as the ANN case. Table 5 lists the learning results of the four GA optimizations. Note that the run-time error norm is much higher than the ANN ones (listed in Table 4); this is because the definition of the run-time error norm in RNN consists of all recurrent cycles. Figure 10 shows that there are 15 recurrent cycles.

The PCA analysis procedure is applied to these four GA optimization listed in Table 5. The performance of the PCA gene, obtained by Equation (6) is listed in the last row of Table 5 and Figure 16. Case 1 in Table 5 does not satisfy the accuracy requirement, but case 2 does. Both cases 1 and 2 use the same GA parameters but different initial GA generation. Hence, the GA optimization results are influenced by the initial GA generation.

On the other hand, one can identify the fast convergence capability of the PCA gene, shown in Figure 16. To decrease the impact of the initial GA generation, it is recommended that one should execute multiple GP optimization processes and then extract these best chromosomes to the PCA gene. Moreover, although RNN provides more information than ANN, both give similar final prediction accuracy, as listed in the last row of Table 4 and Table 5.

### 5.2. The “Progressing” GA Optimizer

The converging behavior of the “progressing” GA optimization demonstrates much difference from the “back-to-original” one. As illustrated in Figure 17, which is the performance of the GA generations under the “progressing” optimizer under an ANN architecture. It clearly depicts the evolving of the chromosomes moving along the backpropagation process. The dashed line of Figure 17 is the convergence curve of the randomly selected 2000 initial GA generation. Comparing the dashes curve to the solid ones, one can identify the continuous optimizing characteristics of the “progressing” GA. The inset of Figure 17 shows the average error norm and size of the generation with respect to the GA generation. Although a continuous convergence of the average error norm can be found, the size of the generation is larger than the “back-to-original” one. This is because the backpropagation algorithm provides various gene combinations with few duplications for the next generation, which might increase the optimization capability of GA with considerable computation resources.

The complete training results are listed in Table 6. All of the denormalized differences satisfy the accuracy requirement of 0.18%. Among all four cases, case 3 performs worse than the other three in terms of the denormalized difference, but it performs well in the run-time error norm. This is because the run-time error norm only reports the learning results of the chromosomes after 500 iterations, but the denormalized difference reflects the 100k learning. This phenomenon depicts the dilemma of selecting how many backpropagation iterations for each GA generation member.

Figure 18 shows the convergence of the four cases and the PCA gene obtained by the same method mentioned above. Although case 4 reveals a low run-time error at the beginning of the learning process, the PCA gene still performs well after approximately 750 learning cycles.

### 5.3. The Impact of the Initial GA Generation

The learning results of the RNN architecture with the “progressing” GA optimizer are shown in Table 7 and Figure 19. Although the overall characteristics, including the convergence curves, the denormalized difference, PCA gene, etc., behave similarly to the previous cases, the overall performance is not good enough, compared to the same RNN architecture with the “back-to-original” GA optimizer, shown in Table 5 and Figure 16.

In order to investigate the impact of the initial GA generation, the ones belonging to Table 5 are applied to the “progressing” GA optimizer, and the learning results are listed in Table 8 and Figure 20. The PCA gene shows fast convergence capability, as illustrated in Figure 20. Comparing Table 7 and Table 8, a clear difference can be identified under the “progressing” GA optimizer with the same GA optimization parameters in terms of run-time error norm and denormalized difference. The influence of the initial GA generation is proven to be significant. Therefore, the recommendation of using the independent initial GA generation for all cases is made.

## 6. Conclusions

In this research, the WLCSP solder joint reliability risk is modeling by an AI model, following the AI-assisted simulation framework. ANN and RNN architectures are conducted to investigate their capability of abstracting the time-dependent solder joint fatigue knowledge from the dataset. The GA optimization is applied to decrease the influence of the initial guessings, including the weightings and bias of the neural network architectures. Due to the continuous learning characteristics of the backpropagation, the “back-to-original” and “progressing” GA optimizers are developed.

Both ANN and RNN architectures, with two hidden layers, are conducted with similar neural network structures. Two GA optimizers are applied to both ANN and RNN architectures with four and six best chromosomes to the next generation. Each GA optimization case starts with an independent 2000 initial GA generation, and each component of the chromosomes follows a zero-mean Gaussian distribution. Moreover, a PCA is applied to the GA optimization results to obtain the PCA gene. PCA gene shows high-speed convergence capability in all cases.

The investigation of the GA optimization shows that increasing the number of the best chromosomes to the next generation and choosing the “progressing” GA optimizer improve the GA optimization results. However, both increase a significant computation resource to conduct. The influence of the initial GA generation is proven to be significant. Therefore, using the independent initial GA generation for all cases and using the PCA gene obtained by several GA optimization processes are recommended.

Because ANN and RNN learnings are more robust due to the GA, these neural networks are suitable for generating response surfaces, as seen in Figure 3g. The predictability of the neural network model enables the exploration of the domain that is outside the training domain (the FEM domain) at a certain range due to the contribution of the nonlinear activation functions [3]. Moreover, due to the continuity of the neural network model, these models are feasible for the optimization procedure.

No matter ANN nor RNN architecture, after 100 k learning iterations, all the AI learning results satisfy the accuracy requirement of 0.18% when the PCA gene is applied as the initial parameter. Moreover, there is no clear evidence that RNN is statistically better than ANN in the WLCSP solder joint reliability risk estimation if the PCA gene is applied. Although RNN provides more information than ANN, RNN is influenced by the noise in the dataset during the learning, which limited the RNN to perform much better than ANN. However, RNN learning requires more computation resources than ANN because of the backpropagation process under a similar neural network structure. Hence, a stable optimization with a broad design domain can be realized by an ANN model with PCA gene with a faster training speed than RNN, even though solder fatigue is a time-dependent mechanical behavior.

## Figures and Tables

**Figure 1 materials-14-04835-f001:**
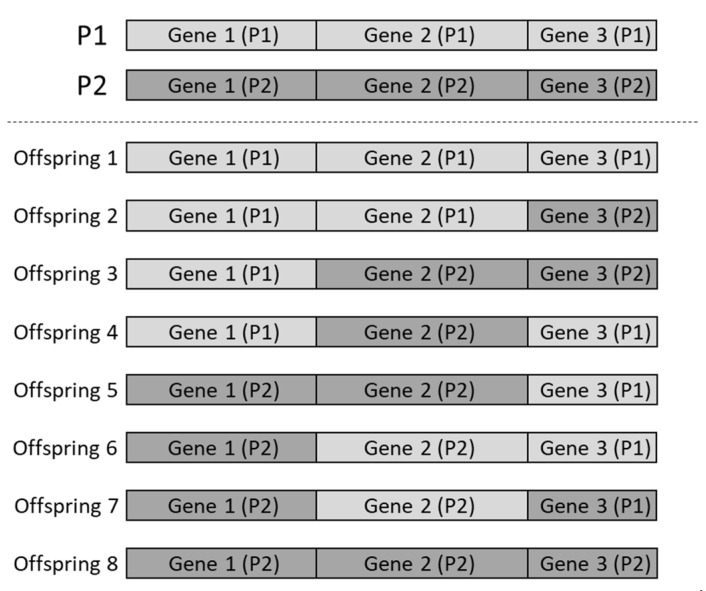
The illustration of the crossover operator.

**Figure 2 materials-14-04835-f002:**
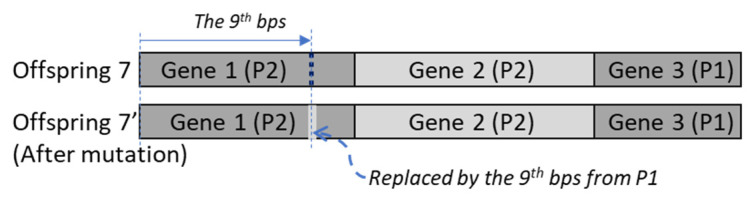
The illustration of the crossover and mutation operator.

**Figure 3 materials-14-04835-f003:**
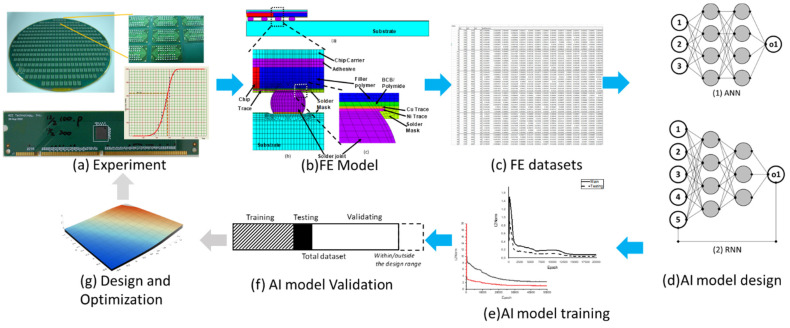
The AI-assisted simulation framework [3,14] (**a**) Experiment; (**b**) FE model; (**c**) FE datasets; (**d**) AI model design; (**e**) AI model training (**f**) AI model validation; (**g**) Design and optimization.

**Figure 4 materials-14-04835-f004:**
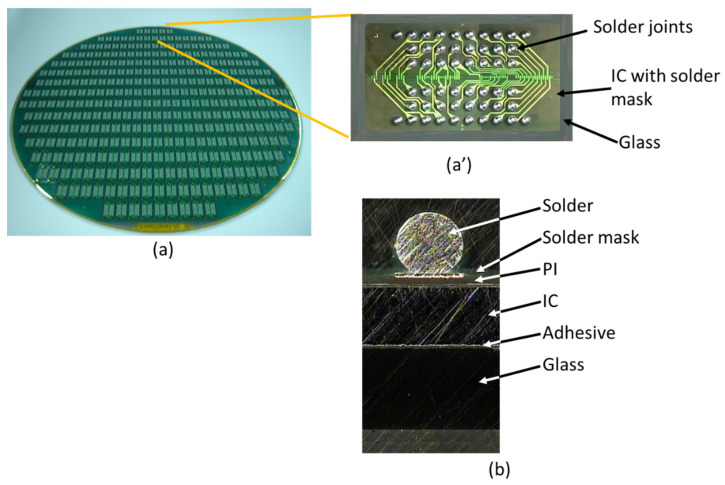
Glass distributed wafer level packaging: (**a**) wafer view with the detail (**a’**) and (**b**) cross-section of a single device [15] (Copyright 2020 EuroSimE).

**Figure 5 materials-14-04835-f005:**
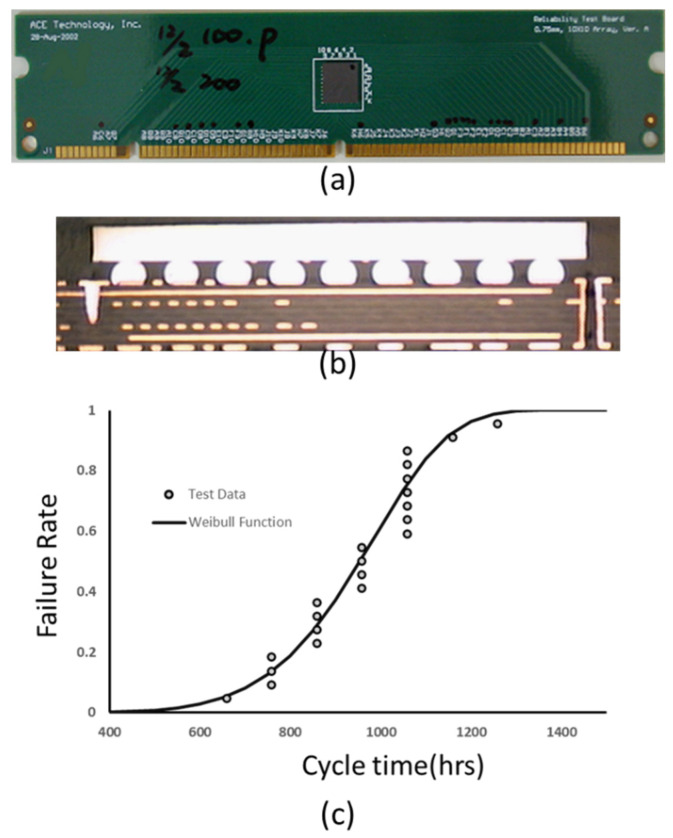
Thermal cycling results: (**a**) the test board, (**b**) the Weibull plot, (**c**) Weibull solder fatigue failure experimental result of 21 samples.

**Figure 6 materials-14-04835-f006:**
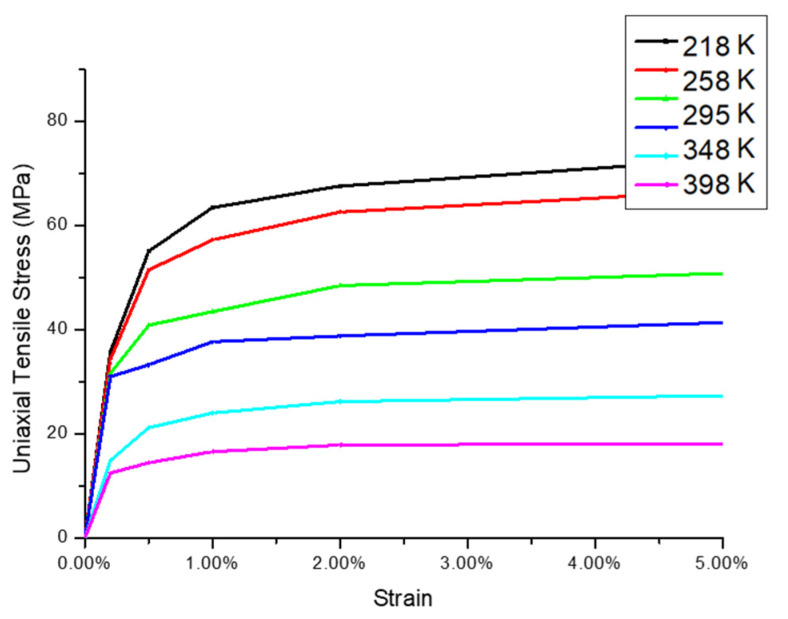
Nonlinear mechanical response of solder joint (63Sn/37Pb) with different temperatures [15] (Copyright 2020 EuroSimE).

**Figure 7 materials-14-04835-f007:**
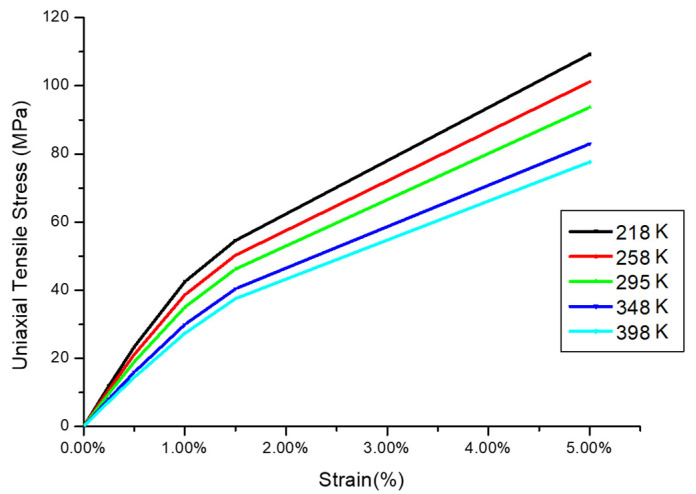
Nonlinear mechanical response of PI with different temperatures [15] (Copyright 2020 EuroSimE).

**Figure 8 materials-14-04835-f008:**
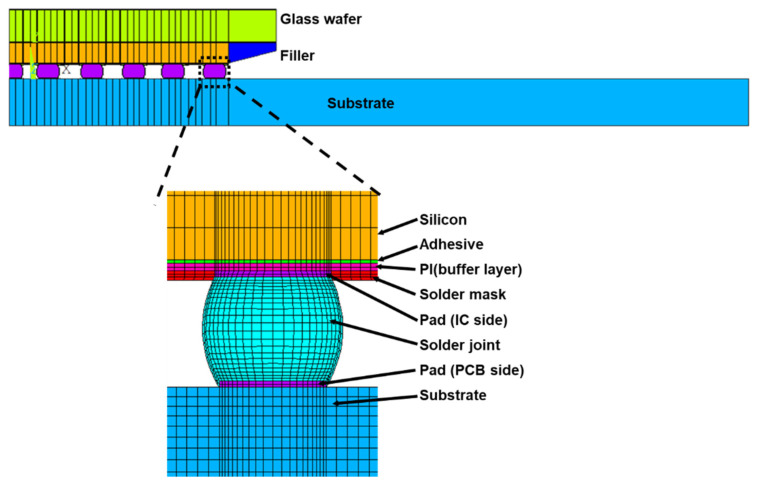
Finite element model for conventional WLCSP and proposed glass WLCSP.

**Figure 9 materials-14-04835-f009:**
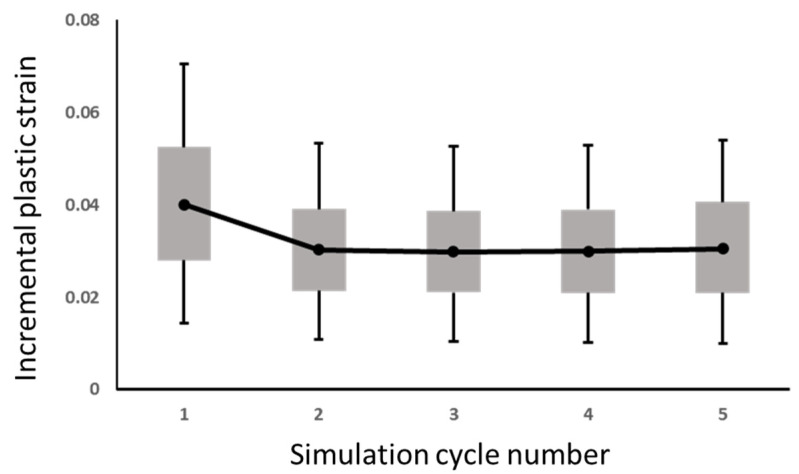
The boxplot of the plastic strain incremental of the 81 data points [15] (Copyright 2020 EuroSimE).

**Figure 10 materials-14-04835-f010:**
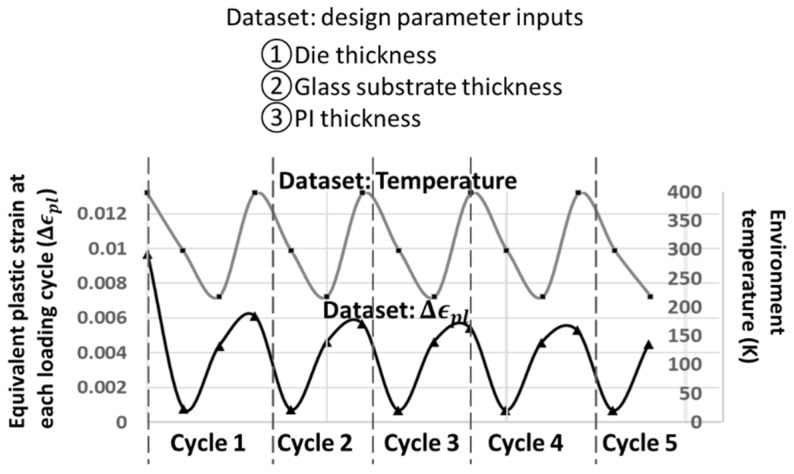
The FE dataset.

**Figure 11 materials-14-04835-f011:**
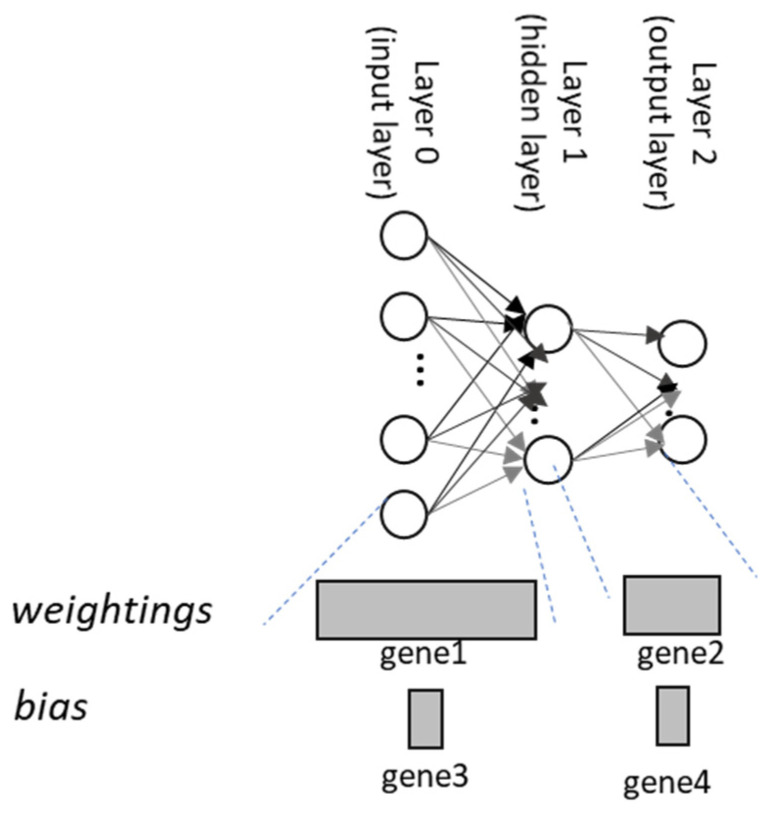
The definition of the chromosomes in GA.

**Figure 12 materials-14-04835-f012:**
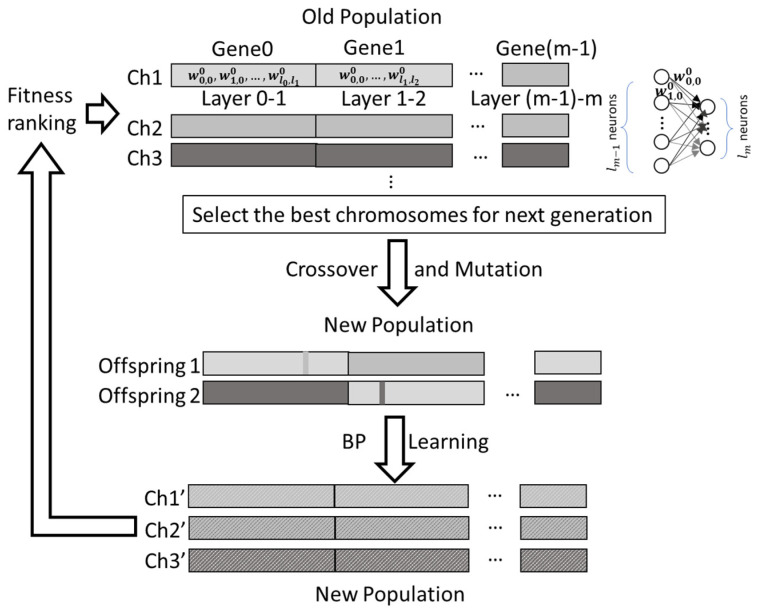
The GA flow chart.

**Figure 13 materials-14-04835-f013:**
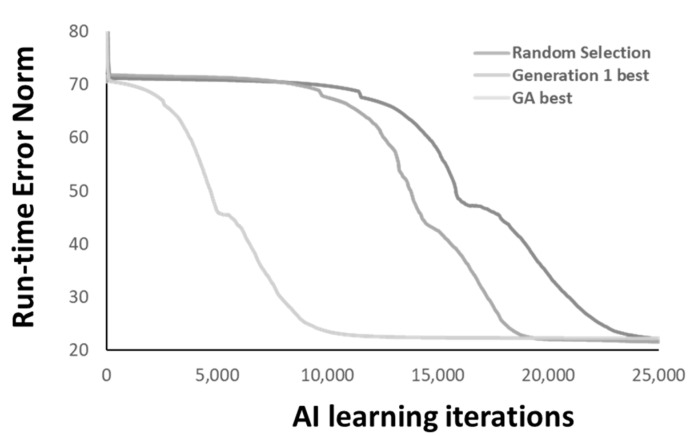
The contribution of GA optimization.

**Figure 14 materials-14-04835-f014:**
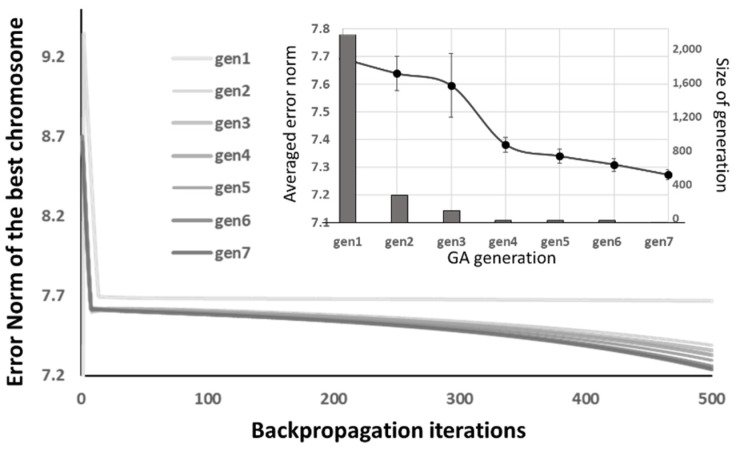
The performance of the GA generations under the “back-to-original” optimizer.

**Figure 15 materials-14-04835-f015:**
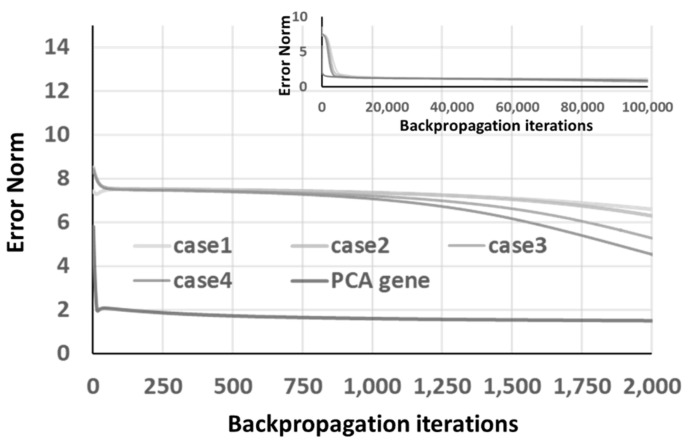
The convergence of 4 GA “back-to-original” optimized chromosomes and PCA gene under ANN architecture.

**Figure 16 materials-14-04835-f016:**
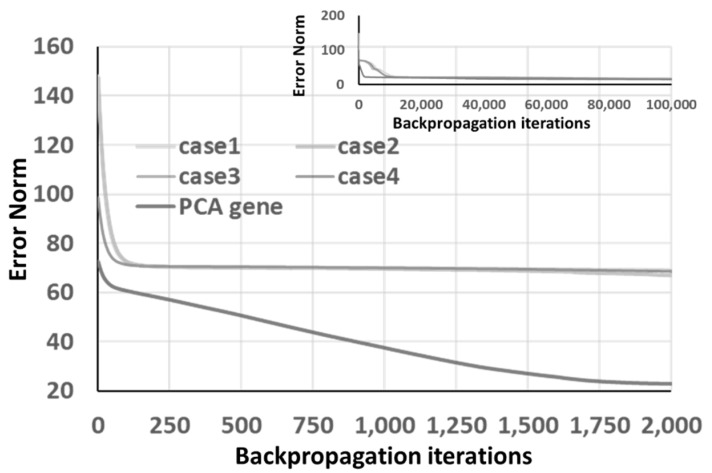
The convergence of 4 GA “back-to-original” optimized chromosomes and PCA gene under RNN architecture.

**Figure 17 materials-14-04835-f017:**
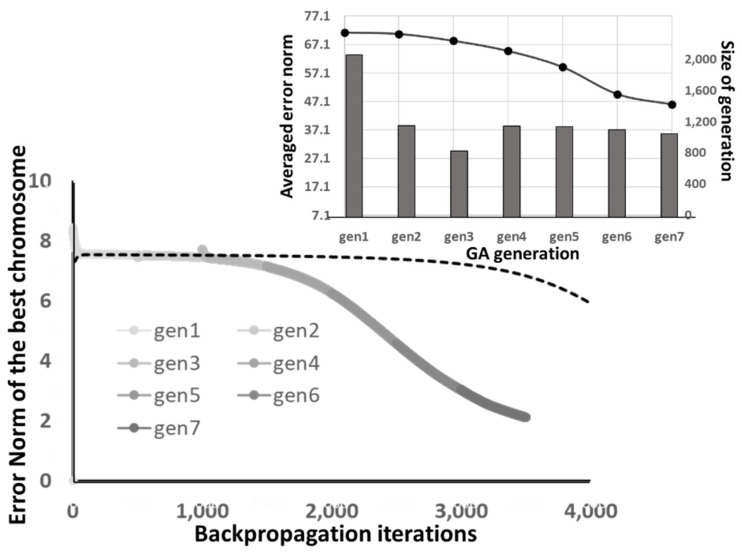
The performance of the GA generations under the “progressing” optimizer.

**Figure 18 materials-14-04835-f018:**
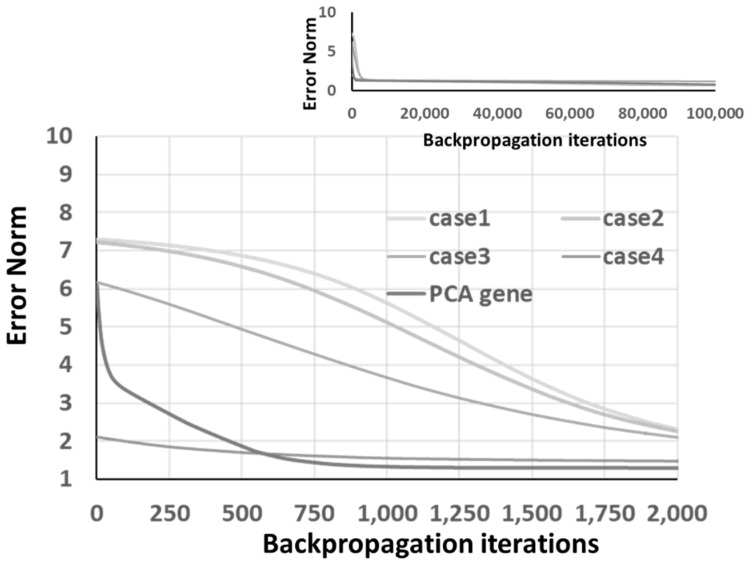
The convergence of 4 GA “progressing” optimized chromosomes and PCA gene under ANN architecture.

**Figure 19 materials-14-04835-f019:**
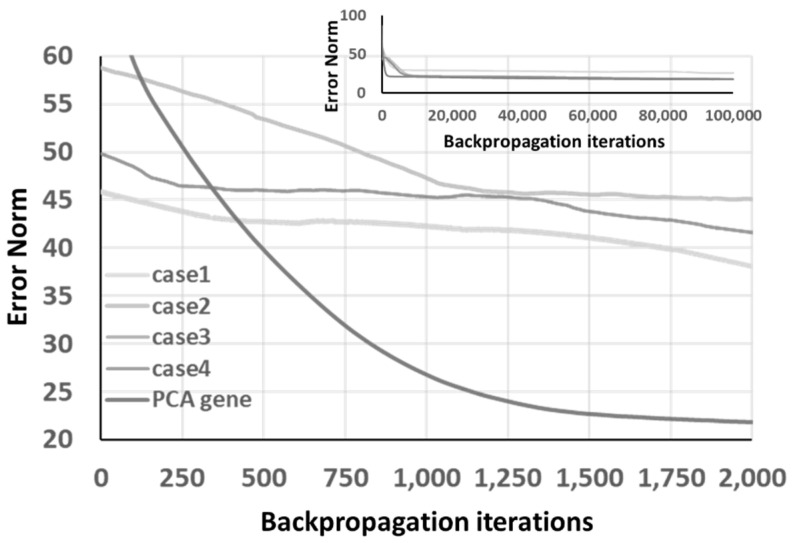
The convergence of 4 GA “progressing” optimized chromosomes and PCA gene under RNN architecture.

**Figure 20 materials-14-04835-f020:**
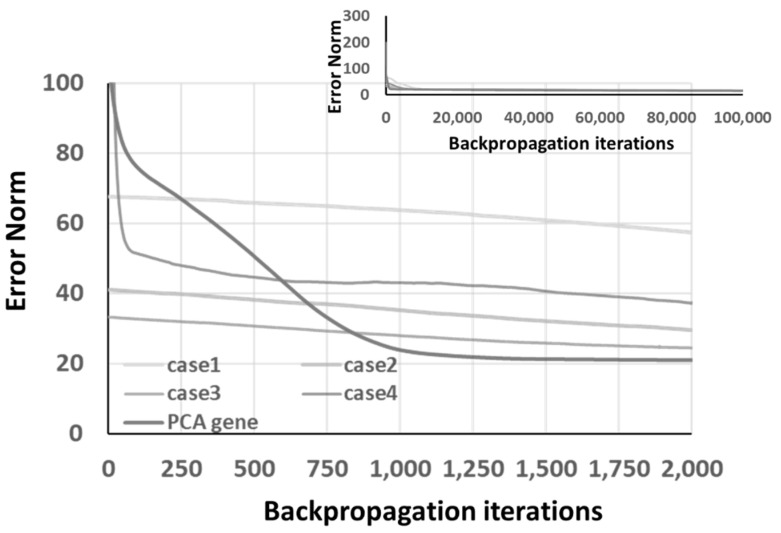
The convergence of 4 GA “progressing” optimized chromosomes and PCA gene under RNN architecture by the same initial 2000 GA generation in Table 5.

**Table 1 materials-14-04835-t001:** Material properties [14].

	Young’s Modules (MPa)	CTE (ppm)	Poission’s Ratio
Solder joint	Temperature dependent and nonlinear (Figure 6)	23.90	0.35
Copper	76,000	17.00	0.35
Solder mask	3400	30.00	0.35
FR4 PCB	18,200	16.00	0.19
PI (Stress Buffer Layer)	Temperature dependent and non-linear (Figure 7)	150.00	0.40
BCB	3000	50.00	0.34
Silicon	112,400	2.62	0.28
Adhesive	0.7	300.00	0.45
Glass	63,000	3.25	0.28
Epoxy	80	250.00	0.34

**Table 2 materials-14-04835-t002:** Finite element model validation.

	Experimental Result (Cycles)	Simulation Prediction (Cycles)
Cycles	1007	1444

**Table 3 materials-14-04835-t003:** Parametric model settings.

Parameter Name	Parameter Alias	Level 1	Level 2	Level 3	Noise Factor Levels
Die thickness	1	0.25 mm	0.375 mm	0.5 mm	±0.015 mm
Glass thickness	2	0.3 mm	0.5 mm	0.8 mm	±0.03 mm
PI thickness	3	0.04 mm	0.025 mm	0.015 mm	±0.005 mm

**Table 4 materials-14-04835-t004:** The ANN training result under GA “back-to-original” optimizer.

	The Best Chromosomes for the Next Generation	GES	Run-Time Error Norm	Denormalized Difference
Case 1	4	6	7.47	0.001530
Case 2	4	6	7.48	0.001287
Case 3	6	7	7.42	0.001225
Case 4	6	7	7.39	0.001098
PCA gene	--	--	1.72	0.001264

**Table 5 materials-14-04835-t005:** The RNN training result under GA “back-to-original” optimizer.

	The Best Chromosomes for the Next Generation	GES	Run-Time Error Norm	Denormalized Difference
Case 1	4	6	69.92	0.002017
Case 2	4	2	70.72	0.001189
Case 3	6	4	69.61	0.001343
Case 4	6	3	70.05	0.001226
PCA gene	--		37.57	0.001090

**Table 6 materials-14-04835-t006:** The ANN training result under GA “progressing” optimizer.

	The Best Chromosomes for the Next Generation	GES	Run-Time Error Norm	Denormalized Difference
Case 1	4	6	6.87	0.001058
Case 2	4	4	6.58	0.001037
Case 3	6	6	4.93	0.001573
Case 4	6	8	1.70	0.001175
PCA gene	--	--	1.87	0.001098

**Table 7 materials-14-04835-t007:** The RNN training result under GA “progressing” optimizer.

	The Best Chromosomes for the Next Generation	GES	Run-Time Error Norm	Denormalized Difference
Case 1	4	8	42.29	0.003281
Case 2	4	6	47.31	0.001663
Case 3	6	6	45.31	0.001592
Case 4	6	7	46.03	0.001254
PCA gene	--	--	26.78	0.001493

**Table 8 materials-14-04835-t008:** The RNN training result under GA “progressing” optimizer by the same initial 2000 GA generation in Table 5.

	The Best Chromosomes for the Next Generation	GES	Run-Time Error Norm	Denormalized Difference
Case 1	4	4	22.70	0.001406
Case 2	4	9	21.46	0.001022
Case 3	6	9	21.78	0.001263
Case 4	6	6	22.00	0.001204
PCA gene	--	--	20.07	0.001110

## Data Availability

The data presented in this research study are available in this article.

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
