# Peer review of "Solder Joint Reliability Risk Estimation by AI-Assisted Simulation Framework with Genetic Algorithm to Optimize the Initial Parameters for AI Models"

_materials, 2021, doi:10.3390/ma14174835_

Round 1

Reviewer 1 Report

This paper presents a framework to develop an AI model to substitute finite element analysis of a solder joint for reliability risk estimation. The topic of this paper is interesting and this paper well summarizes the authors' work.  However, there are some weak point that should be improved. 

  1. The procedure for finite element analysis of solder joint model is well explained but the reason why you select "the thicknesses" as important model parameter for reliability risk estimation should be addressed.
  2. In Chapter 5, What is the role of GA optimization? Is the training set for AI training is generated using GA optimization? How many sets are generated from GA optimization?
  3. In figure 13, what does the y-axis mean?   
  4. It seems that the resulting AI model can be used for design process. Could you please add schematic explanation about how to use the model after training and include comparison validation result with the FEA result? 

Reviewer 2 Report

The paper presents the estimation of solder joint reliability risk by AI-assisted simulation framework with genetic algorithm the goal in to optimize the initial parameters for the AI models. After careful review, I recommend to accept the paper with minor revision.

The following are the comments/suggestions on the paper:

  • Please check MDPI guidelines on writing the keywords in the Keyword section
  • In section 2. Theory, please provide more information of the resources and provide appropriate reference/references
  • In section 3. The AI-assisted simulation framework and FE datasets preparation, explain about the parametrization for better understanding.
  • I have attached the similarity index for the manuscript. Please check the word-to-word similarity in section 3. The AI-assisted simulation framework and FE datasets preparation. I recommend rewriting these sections with the original work presented in the manuscript.

Reviewer 3 Report

Thanks to submit the manuscript entitled “Solder joint reliability risk estimation by AI-assisted simulation framework with genetic algorithm to optimize the initial parameters for AI models”. The manuscript is prepared in a standard form. However, some information of the present work needs more clarification and more details. Hence, a major revision is needed. The detailed comments are described below:

  1. The presented results in in the abstract or conclusion are not enough clear.
  2. It is suggested to rephrase lines #49 and #50 to avoid any misunderstanding that how AI can help to minimize the use of FE models, as it can generate the paradigm for a generic route.
  3. Some typo error such as p 2. Line 95, “AL model training with GA optimized ..." should be corrected.
  4. What do you mean with "static thermal cyclic loadings" as mentioned in P.9 Line 220?
  5. Section 2, Theory, presents the background of theory. However, it does not include the references to understand how and where the notes are gathered. The reviewer also suggests presenting the theory by using an example related to the electronic package case (if it is possible).
  6. The neural networks (ANN or RNN) are used and the questions is that what it offers more than what other traditional machine learning algorithms (such as CNN) can offer?
  7. Can the authors please clarify 2D or 3D simulation is implemented for this work? If, 2D simulation is used, can you please explain that how the results for modeling a 3D geometry (bulb joint) are validated?
  8. Are the experiment and FE model simulations shown in Figure 3, implemented by the authors? If no, please also provide the reference.
  9. Line 161, Figure 4; the figure caption contains (c) schematic cross-section. However, the figure does not show the view of C (Figure 4 (c)?).
  10. The quality of Figure 5 (c) needs to be improved. Also, there is no unit for the Time axis (might be hrs.). The reviewer also asks the authors to explain the relationship between time and the mentioned fatigue cycle number (from the experiment) in line #172 (1447 cycles). Also, according to Table 2, 1444 cycles is the fatigue cycle number estimated by the simulation, and 1007 cycles are from experiments. Please clarify the difference between the experimental results (1007 and 1447).
  11. Please add the reference for the data used in Figure 6, Figure 7, and table 1 to their captions.
  12. Please mention the creep parameters used in the analysis and also mention the related reference.
  13. Can you please explain that how the number of chromosomes are determined for ANN and BNN?

Kind Regards

Round 2

Reviewer 1 Report

In revision, the authors have addressed all points I made. One suggestion for better legibility is that some figures need to be have higher resolution.

Reviewer 3 Report

I appreciate your responses to the comments and your concern about updating the manuscript.
The updated manuscript is much more suitable to be published in the journal of Materials. However, A minor revision shall be required. For this, the authors have been asked to consider the below comment in the last version of the paper.
Using the 2D simulation for the 3D bulb solder joint led to errors in the results as the aspect ratio of the bulb joint is not enough high (due to the almost spherical geometry of the bulb joint). Also, in the present work, the time-dependent material property (creep behaviour) is not considered for the solder, and this assumption causes another error in the FEM results. These two recent errors are probably the main reasons for the difference between the experimental and numerical results (about 40%).
So, the authors have been asked to report the reasons for the errors in the results provided in the manuscript.

Thank you very much again for your attention to publish your work.

Kind Regards
